# Nabilone treatment for severe behavioral problems in adults with intellectual and developmental disabilities: Protocol for a phase I open-label clinical trial

Hsiang-Yuan Lin [1,2,3]*, Elia Abi-Jaoude[3,4], Pushpal Desarkar[1,2,3,5], Wei Wang[3,6], Stephanie H. Ameis[1,3,7,8], Meng-Chuan Lai [1,3,4,7,8,9,10], Yona Lunsky[1,3], Tarek K. Rajji [2,3,11,12]

1 Azrieli Adult Neurodevelopmental Centre, Campbell Family Mental Health Research Institute, Centre for Addiction and Mental Health, Toronto, Ontario, Canada, 2 Adult Neurodevelopment and Geriatric Psychiatry Division, Centre for Addiction and Mental Health, Toronto, Canada, 3 Department of Psychiatry, Temerty Faculty of Medicine, University of Toronto, Toronto, Ontario, Canada, 4 Department of Psychiatry, The Hospital for Sick Children, Toronto, Ontario, Canada, 5 Temerty Centre for Therapeutic Brain Intervention, Centre for Addiction and Mental Health, Toronto, Ontario, Canada, 6 Centre for Complex Interventions, Campbell Family Mental Health Research Institute, Centre for Addiction and Mental Health, Toronto, Ontario, Canada, 7 Margaret and Wallace McCain Centre for Child, Youth & Family Mental Health, Campbell Family Mental Health Research Institute, Centre for Addiction and Mental Health, Toronto, Canada, 8 Program in Neurosciences & Mental Health, Sick Kids Research Institute, The Hospital for Sick Children, Toronto, Canada, 9 Department of Psychiatry, Autism Research Centre, University of Cambridge, Cambridge, United Kingdom, 10 Department of Psychiatry, National Taiwan University Hospital and College of Medicine, Taipei, Taiwan, 11 Campbell Family Mental Health Research Institute, Centre for Addiction and Mental Health, Toronto, Ontario, Canada, 12 Toronto Dementia Research Alliance, University of Toronto, Toronto, Ontario, Canada

* Hsiang-Yuan.Lin@camh.ca

**Data Availability Statement:** No datasets were generated or analysed during the current study. All

## Abstract

Severe behavioral problems (SBPs) are common contributors to morbidity and reduced quality of life for adults with intellectual and developmental disabilities (IDD) and their families. Current medications for SBPs show equivocal effectiveness and are associated with a high risk of side effects. New and safe treatments are urgently needed. While preliminary studies suggest that medical cannabinoids, particularly the synthetic cannabinoid nabilone, are plausible treatment options for SBPs in adults with IDD, data on the tolerability, safety and efficacy of nabilone in this population has never been investigated. Thus, we propose this first-ever Phase I pre-pilot open-label clinical trial to obtain preliminary data on the adherence, tolerability and safety profiles of nabilone in adults with IDD, and explore changes in SBPs pre- to post-treatment. We hypothesize that nabilone has favorable tolerability and safety profile for adults with IDD. The preliminary results will inform the next-stage pilot randomized controlled trials, followed by fully powered clinical trials eventually. This research helps fill the evidence gap in the use of cannabinoids in individuals with IDD to meet the needs of patients, families, and service providers.

relevant data from this study will be made available upon study completion.

**Funding:** This project is funded by the CAMH Innovation Fund of the Alternative Funding Plan for the Academic Health Sciences Centres of Ontario, and the University of Toronto Department of Psychiatry Excellence Funds. H-YL is supported by the Azrieli Adult Neurodevelopmental Centre at Centre for Addiction and Mental Health, and an Academic Scholar Award from the Department of Psychiatry, University of Toronto. The funders had no role in the design of the study; in the collection, analyses, or interpretation of data; in the writing of the manuscript, or in the decision to publish the protocol.

**Competing interests:** The authors have declared that no competing interests exist.

# Introduction

## Intellectual and developmental disabilities and aggression

Approximately 1–3% of adults across the world have some form of intellectual and/or developmental disabilities (IDD) [1, 2], and 20–40% of them experience severe behavioral problems (SBPs), including self-injurious behavior, disruptive behavior, or aggression [3, 4], in the absence of a psychiatric diagnosis. These SBPs are associated with emotion dysregulation, pain, and sleep problems [5]. SBPs are major contributors to morbidity, functional impairments, missed opportunities for learning, reduced quality of life, and well-being challenges on families, health, education and disability sectors [6].

Although environmental adjustments and psychosocial interventions are considered first-line treatments for SBPs in IDD based on national and international guidelines [7, 8], these options are not always effective. Antipsychotics and other psychotropic medications are thus prescribed for up to 70% of adults with IDD [9, 10], despite equivocal efficacy, and many of them are used in an off-label way [11]. People with IDD, relative to those without IDD, are at a higher risk of side effects of psychotropic agents and are less able to report side effects [12, 13]. These side effects include weight gain, metabolic disturbances, neurological symptoms, and avoidable death [14]. Furthermore, because of suboptimal responses to currently available medications, polypharmacy is commonly used for SBPs in adults with IDD [10], potentially leading to more adverse events [11]. Thus, new and safer interventions are pressingly needed for this population.

## Endocannabinoid system and aggression

Neurophysiologically, the endocannabinoid system may be involved in SBPs and aggression through a complex interplay with other systems [15]. Specifically, stimulated by Δ9-tetrahydrocannabinol (THC), the primary psychoactive ingredient in cannabis, the cannabinoid receptor 1 (CB1) modulates the activity of serotonergic, noradrenergic, dopaminergic, γ-aminobutyric acid (GABA)ergic, and glutamatergic neurotransmitters. CB1 is a potent modulator of myriads of cognitive and behavioral processes including learning, attention, emotion regulation, sleep structure, and pain perception [16]. The cannabinoid receptor 2 (CB2), also activated by THC, is mainly localized in peripheral immune cells and is expressed in the brain during neuroinflammation. It has been postulated that neuroinflammatory processes alter the metabolism and functions of neurotransmitters, neuroendocrine activity and neural plasticity, which could lead to the development of neuropsychiatric symptoms [17]. Neuroinflammatory markers have close connections with peripheral inflammatory cytokines [18], whose levels correlate with aggression in children with Prader-Willi Syndrome (60% of these children have IDD and those without IDD have some form of learning disabilities) [19], healthy adults [20], and people with intermittent explosive disorder [21]. Further, CB1 [22, 23] or CB2 [24] knockout mice show increased aggression compared to wild-type mice. In this context, CB2 agonist appears to provide a neuroprotective effect on neuroinflammation [25]. CB1 [26] and CB2 [24] agonists at lower doses have been shown to reduce aggression in preclinical studies. A theoretical review also suggests that CB1 modulation may be a future candidate for the treatment of self-injury [27].

Human functional MRI (fMRI) data suggest that at low doses, THC attenuates anxiety responses and reduces amygdala reactivity and its functional coupling with frontal regions during processing of stimuli with negative emotional content. Conversely, at higher doses, the effect of acute administration of THC becomes anxiogenic [28]. In parallel, human proton magnetic resonance spectroscopy ($^1$H MRS) studies indicate that the GABA and glutamate-glutamine balance, which is modulated by CB1, is associated with emotion regulation and related psychopathology [29]. Taken together, this evidence suggests that agents which can

activate CB1 and CB2 may help decrease SBPs and aggression in adults with IDD given the essential roles of endocannabinoid and neuroimmune systems in a multitude of behavioral functions associated with aggression.

## Why nabilone?

Anecdotal reports [30] and preliminary research [31] suggest that medical cannabinoids may be useful to treat SBPs and aggression in individuals with IDD. However, despite high hopes held by families [32], the scientific evidence is still in a very early stage. Most studies to date focused on cannabidiol (CBD), as THC use has been associated with cognitive decrements, psychotic symptoms, and addiction in recreational marijuana users [33]. However, the mechanism of action and appropriate dose range of CBD, which, unlike THC, has only a very low affinity for CB1 and CB2, remains elusive. Moreover, prior evidence suggests limited beneficial properties of CBD for aggression and risk factors associated with SBPs [34]. Another option is the plant form of medical cannabis. However, individuals have variable sensitivity to different cannabis strains with varied combinations of THC, CBD, and other ingredients [35], making it difficult to investigate the effects of plant-form cannabis in a controlled trial setting. Therefore, identifying a cannabis derivative with a clearer mechanism and applicability for SBPs in IDD is needed.

Nabilone is a synthetic oral THC analogue that acts as a partial agonist on both CB1 and CB2 in humans. Thus, it mimics the effect of THC on aggression, emotion regulation, sleep, and pain, but with more predictable side effects, better safety profiles, and less euphoria [36]. Clinical studies suggest that nabilone may be useful in alleviating agitation in patients with dementia [37, 38], nightmares in patients with post-traumatic stress disorder (PTSD) [39], non-motor symptoms (i.e., mood dysregulation, sleep, pain) in patients with Parkinson's disease [40], and core symptoms in those with anxiety and pain disorders [39]. Preliminary studies suggest that another synthetic THC, dronabinol (a full agonist at CB 1 and CB2), may be effective in alleviating SBPs in youth with IDD [41, 42]. However, dronabinol has been withdrawn from the Canadian market for unknown reasons. Moreover, nabilone, relative to dronabinol, has better bioavailability [43] and may be more effective in alleviating SBPs [37]. Notably, nabilone may have fewer adverse mental health effects such as agitation, irritability and psychosis, because, like THC, it has relatively weak partial agonist activity at CB1 while most other synthetic cannabinoids (including dronabinol) exhibit full CB1 agonist properties [15, 44, 45].

Nabilone is indicated for severe nausea and vomiting associated with cancer chemotherapy and is commercially available in Canada. Its use is usually safe and satisfactorily tolerated. The most frequently reported adverse reactions in previous clinical trials are drowsiness, vertigo, mild psychological high, and dry mouth, depending on the dose and population [36–38, 40, 46, 47]. The safety profile and rate of adverse events for nabilone are comparable to those reported in studies of CBD [48] and clonidine [49], and appear more favorable than what has been reported for antipsychotics [11, 13, 14]. Importantly, the addictive potential of nabilone is very low [50] because it is associated with less euphoria, slower onset of action, and more difficult titration compared to smoking cannabis. In fact, nabilone has been shown to help with sustained abstinence from marijuana [51]. The incidence of nabilone-induced psychotic symptoms is rare and has only been reported in people with a personal or family history of psychosis.

## Study objective

Based on the encouraging evidence in patients with dementia [37] and Parkinson's disease [40] as well as anecdotal clinical observations, nabilone could be a promising and novel

treatment for SBPs in adults with IDD. However, the efficacy and safety profiles of nabilone in this population have yet to be studied. To enhance the current state of evidence for nabilone use to treat SBPs and aggression in adults with IDD, we propose to conduct the first phase I open-label clinical trial with the primary objective being to collect preliminary data on the tolerability and safety profile of nabilone in adults with IDD. To evaluate tolerability, we hypothesize that 80% of participants will complete the open-label treatment stage. To evaluate safety, we hypothesize that nabilone will be satisfactorily tolerated and have a favorable safety profile in adults with IDD, as quantified by lower rates of adverse events than those reported in the literature of antipsychotics treatment for SBPs in individuals with IDD. The explorative objective is to describe changes in SBPs and aggression in participants pre- to post-treatment.

## Materials and methods

### Trial design

This study is a Phase I pre-pilot open-label trial of nabilone in adults with IDD and SBPs. As shown in Figs 1 and 2, the trial first includes a screening visit (S-V) for eligibility, then a baseline visit (V 0), followed by a dose-titration phase, and then a 4-week open-label phase at a stable dose. At the end of the open-label phase, participants will undergo a termination visit (V 1), after which nabilone will be tapered off over 8 days. Subsequently, 2 weeks after the full discontinuation of nabilone, participants will undergo the last follow-up visit (V-F) to ensure safety.

### Procedures

Participants will be recruited through the Adult Neurodevelopmental Services at the Centre for Addiction and Mental Health (CAMH) in Toronto, Canada, or from the community (self-referral). Advertisements will be posted on CAMH Research Registry, social media channels, and community organizations providing services for individuals with IDD and their families.

**Screening visit (S-V).** Potential participants and their substitute decision-makers (SDMs) will attend a research screening visit. Before the informed consent process, the participant will be assessed for capacity to provide informed consent (see S1 File for the Informed Consent Form). If he/she/they cannot pass the competence test using the MacArthur Competence Assessment Tool for Clinical Research (MacCAT-CR) [52], which provides a structured format for capacity assessment, then the written informed consent will be obtained from their SDM after seeking the participant's assent (see S2 File for the Assent Form). A score of 70% or higher on the MacCAT-CR will indicate that a given participant is deemed competent to consent, given no relevant standard for people with IDD in the prior literature. This cut-off is in between the cut-offs used for adults with schizophrenia [53] and neurotypical children [54], which we deem to be a middle ground to balance sensitivity and specificity in assessing capacity to consent. In addition to the MacCAT-CR, the research team will also utilize clinical assessments of the potential participants to determine their capacity to consent to participate in this research, in a similar manner to determining capacity for clinical treatments as outlined in clinical guidelines [55]. Participants will then be assessed for eligibility based on the inclusion and exclusion criteria (described below). We will also assess participants' other psychiatric conditions using the Moss Psychiatric Assessment Schedules (The Moss-PAS (ID), previously called Mini PAS-ADD) [56], and baseline adaptive function using the caregiver-rated Adaptive Behavior Assessment System-III (ABAS-3), Adult Form [57].

**Baseline visit (V 0).** At V 0 the following assessments will be completed:

| | Screening (S-V) | Pre-Treatment (V 0) | Titration | Open-label | Post-Treatment (V 1) | Safety Follow-up (V-F) | Visit option |
|---|---|---|---|---|---|---|---|
| Informed Consent | • | | | | | | |
| **Demographics** | | | | | | | |
| Dx of IDD and comorbidities (Moss-PAS (ID)) | • | | | | | | Either or[a] |
| Age | • | | | | | | Either or |
| Sex and Gender | • | | | | | | Either or |
| Height | • | | | | | | In person |
| Weight | • | • | | | • | • | In person |
| Competence (MacCAT-CR) | • | | | | | | Either or |
| Concomitant medication | • | • | • | • | • | • | Either or |
| Adaptive function (ABAS-3) | • | | | | | | Either or |
| Adherence | | | | | • | | Either or |
| **Safety Outcome** | | | | | | | |
| PE, Vital signs | • | • | | | • | • | In-person |
| Pregnancy test | • | | | | | | In-person |
| AEs (UKU) | | | • | • | • | • | Either or |
| NIH-Toolbox | | • | | | • | • | In-person |
| **Clinical Effect Outcome** | | | | | | | |
| *Primary* | | | | | | | |
| ABC-I | • | • | | | • | | Either or |
| *Secondary* | | | | | | | |
| MOAS | | • | | | • | | Either or |
| Anxiety-ADAMS | | • | | | • | | Either or |
| Stress-DASS-21 | | • | | | • | | Either or |
| BFDS | | • | | | • | | Either or |
| CGI-S | • | | | | • | | Either or |
| CGI-I | | • | | | • | | Either or |
| *Exploratory* | | | | | | | |
| SCQ-AIQ | | • | | | | | Either or |
| MRI (optional) | | • | | | • | | In-person |
| **Nabilone Dispense** | | | | | | | |
| Pick-up | | • | | • | • | | In-person |
| Return (if applicable) | | | | • | • | • | In-person |
| **Acceptability Evaluation** | | | | | | • | Either or |

**Fig 1. Schedule of study visit procedures and assessments.** Abbreviation: Dx = diagnosis; MacCAT-CR = MacArthur Competence Assessment Tool for Clinical Research; Moss-PAS (ID) = Moss Psychiatric Assessment Schedules; ABAS-3 = Adaptive Behavior Assessment System, Third Edition; PE = physical examination; AE = adverse event and severe adverse event; NIH-Toolbox = NIH Toolbox® Cognition Battery; ABC-I = Aberrant Behavior Checklist-Irritability Subscale; MOAS = Modified Overt Aggression Scale; CGI-I = Clinical Global Impressions (CGI) scale-Improvement; CGI-S = Clinical Global Impressions (CGI) scale-Severity; Anxiety-ADAMS = General Anxiety Subscale of Anxiety, Depression, and Mood Scale; Stress-DASS-21 = Stress Subscale of the short-form version of Depression Anxiety Stress Scales; BFDS = Brief Family Distress Scale; SCQ-AIQ = Social Communication Questionnaire for Adults with Intellectual Disability. [a]: Either in-person or virtual visit.

- Clinical Global Impression–Severity scale (CGI-S) [58] will be rated by the investigator to evaluate the severity of SBPs.

- Baseline vital signs and weight.

- Participants' cognitive capacity using the NIH Toolbox® Cognition Battery (NIHTB-CB). Previous evidence suggests that nabilone may cause acute mild decrement in attention and working memory [37]. We will use the NIHTB-CB, which has been shown as a good candidate measure to examine broad nonverbal cognitive changes for individuals with IDD (valid and reliable for those with a mental age of ≥5 years; implementable for those with a mental age of ≥3 years) [59], to assess the baseline attention and working memory.

- SBPs, including behavioral problems using the Aberrant Behavioral Checklist-Irritability subscale (ABC-I) [60, 61] and Modified Overt Aggression Scale (MOAS) [62], as reported by the caregivers. These scales will be the principal measures for the detecting changes in SBPs.

- Anxiety, using the parent-rated General Anxiety Subscale of the Anxiety, Depression, and Mood Scale (ADAMS) [63]. Nabilone and THC at low doses have been shown to alleviate anxiety [39]. Hence, this measure will be used for exploratory aims to detect a signal of changes in anxiety with nabilone use.

- Caregiver stress and crisis using the Stress Subscale of Depression Anxiety Stress Scales (DASS-21) [64] and Brief Family Distress Scale (BFDS) [65]. As SBPs often poses major

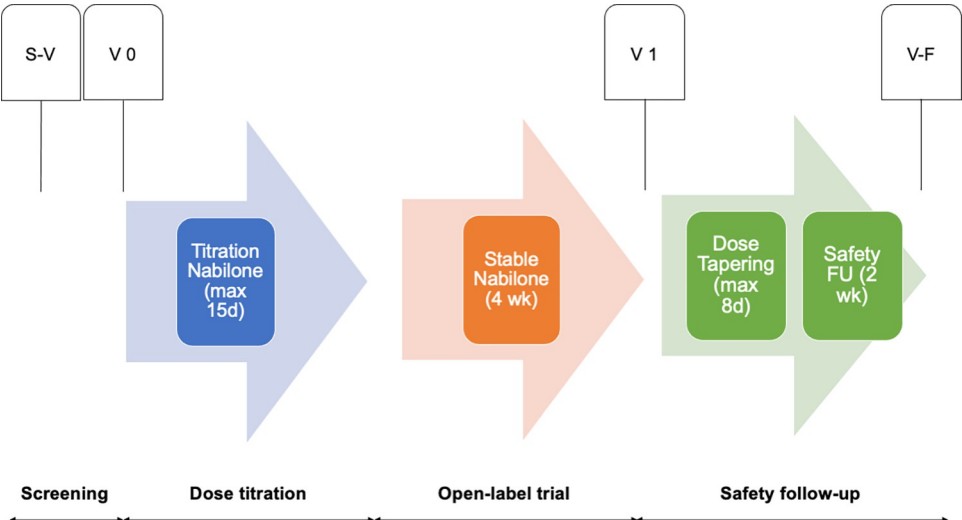

**Fig 2. A schematic diagram of the trial design, including all study visits and timeline.** Abbreviation: S-V: screening visit; V 0: baseline visit; V 1: visit at the end of open-label phase; V-F: follow-up visit to ensure safety; d: days; wk: weeks; FU: follow-up; max: maximum.

well-being challenges for caregivers, we aim to assess whether caregiver stress and distress improve if nabilone also improves SBPs of the participant.

- Autistic characteristics using the Social Communication Questionnaire for Adults with Intellectual Disability (SCQ-AID) [66]. Prior evidence suggests that autistic characteristics may be associated with SBPs [67]. We aim to explore whether baseline autistic characteristics modulate the treatment effects of nabilone.

- MRI Imaging (optional, dependent on the participant's capability to complete the scans, to be determined based on the opinions of the caregivers/parents and investigator).

- Concomitant medication(s).

**Titration and open-label phases of nabilone treatment.** After V 0, eligible participants will receive open-label nabilone starting with a dosage of 0.25 mg at bedtime. During the dose titration phase (which could last up to 15 days), nabilone will be titrated in 0.25 mg increments every two days, using a twice daily dosing schedule, after consultation with the study team during regular phone calls. This proposed up-titration schedule was decided based on a clinical trial of nabilone treatment for non-motor symptoms in adults with Parkinson's disease [40]. These regular phone calls will be carried out every two days during the whole dose titration phase until the participant reaches a stable dose for the open-label phase. The study team will also check whether the participant shows any physical or mental changes believed to be attributed to adverse events based on the UKU side effect rating scale [68], and advise the participant and his/her/their caregivers accordingly on the next step of titration. Dose adjustments are performed until the participant reaches the maximum permitted dose of 1 mg twice daily or experiences intolerable adverse events believed to be related to nabilone. If intolerable, the participant will use nabilone at the previous lower dose, entering the open-label phase. Note that the suggested maximum nabilone daily dose per the product monograph is 6 mg. However, considering the potential vulnerability of individuals with IDD, we chose to administer a maximum total of 2 mg/day in the current proposal, identical to the maximum dose used in clinical trials in patients with dementia [37] and Parkinson's disease [40]. Moreover, nabilone is an analog of THC, which appears to reduce anxiety at low doses but becomes anxiogenic at high doses [69].

Participants will then be on a stable, optimized nabilone dosage for the four-week open-label phase, after which the trial ends with an on-site termination visit (V 1). During the open-label trial phase, the study team will call the participant and their caregivers once weekly to perform a basic check-up and to follow up with any questions or concerns related to the trial. If there are any changes suspected to be adverse events, the UKU side effect rating scale will be conducted during the telephone check-up. Caregivers will use the drug diary to record each administration of nabilone, including time, dosage, concomitant medications, and any noteworthy comments.

**Termination (post-treatment) visit (V 1).** At V 1, the following post-treatment assessments will be completed:

- Adverse events using the UKU side effect rating scale, conducted by the investigator, to systemically investigate physical adverse events of nabilone. The UKU side effect rating scale is a short and easy-to-use instrument that captures the core dimensions of side effects in patients using psychotropic medications and is delivered by interviewing and observing the patient and their caregivers. In this study, we will adopt the UKU side effect rating scale specifically adjusted to adults with IDD, which consists of 35 out of the original 48 items [68].

- Vital signs (including blood pressure to check orthostatic hypotension) and weight to monitor the physical adverse events of nabilone.

- Participants' cognitive capacity using the NIH Toolbox® Cognition Battery, to monitor any acute cognitive change (considered as an adverse event) due to nabilone.

- Caregiver-rated ABC-I, MOAS, and General Anxiety Subscale of ADAMS. CGI-S and CGI-I will be rated by the investigator to evaluate the severity and improvement, respectively, of the SBPs after nabilone treatment. These re-assessments are implemented for the exploratory study aim of assessing behavioral changes pre- to post-treatment of nabilone.

- Caregiver's stress and crisis using the Stress Subscale of DASS-21 and BFDS, to investigate changes in caregiver's stress and distress after the participant's nabilone treatment.

- Optional MRI Imaging for those who complete the pre-treatment MRI and are willing and able to undergo a second of MRI scan. This is to explore changes in brain function and structure pre- to post-treatment of nabilone.

**Tapering phase and safety follow-up visit (S-F).** Nabilone will be tapered in 0.25 mg daily decrements to prevent acute withdrawal effects. This down-titration schedule was decided based on a conservative modification from a clinical trial using nabilone to treat behavioural and psychological symptoms of dementia [37]. Phone calls will be held every other day during the dose-tapering phase to check whether there are any changes believed to be attributed to withdrawal effects. At the completion of the tapering of nabilone, there will be a final reconciliation of the nabilone dispensed, consumed and remaining, based on the drug diary and the remaining drug doses returned. This reconciliation will be logged on an accountability form and used to measure compliance.

A safety follow-up visit (S-F) will be scheduled after 2 weeks of full discontinuation of the study drug. During the S-F, vital signs, the UKU side effect rating scale and NIH Toolbox® will be assessed again to check for any residual adverse event. During the S-F, participants (if capable) and caregivers will complete a custom-designed feedback questionnaire (S3 File) regardingthe acceptability of all study components, including recruitment, withdrawing from the study, study visit attendance, protocol adherence and the time burden of completing questionnaires. This caregiver and participant perspective will guide the planning for a subsequent clinical trial for this population [70]. This practice will help us understand barriers and how best to facilitate participation in medication trials for participants with IDD and their families.

## Elective virtual assessment

To reduce barriers associated with traffic and time commitment, and to future-proof the research protocol in case of another lockdown for next pandemic, we will provide the option for virtual assessments. As shown in the last column of Fig 1, interviews that collect data on demographics, clinical history, diagnoses alongside medication use, competence, and adverse effects can be conducted virtually using teleconferencing software. Additionally, a blank paper copy of outcome measures rated by caregivers (including the ABC-I, MOAS, Anxiety-ADAMS, Stress-DASS-21, BFDS, and SCQ-AIQ) can be sent to them via mail upon request. Completed questionnaires can be mailed, faxed, scanned, or photographed back to the study team. If the caregivers choose to send back the completed questionnaire by mail, we will provide a pre-paid postage envelope.

The virtual sessions will be undertaken in coordination with in-person sessions based on the same research standards. The participants and caregivers can flexibly choose either the in-

person or virtual assessments where possible (Fig 1). We will notify both the participants and their caregivers of this virtual option when explaining the study and discussing the informed consent.

## Outcome measures

The primary outcome measures are adherence (as quantified by counting pill usage), adverse events, and Crystallized and Fluid Composite scores of the NIH Toolbox® Cognition Battery [59]. An adverse event will be defined as any untoward medical occurrence in a participant, temporally associated with his/her/their involvement in this study (from the time that the informed consent document is signed to the S-F), without regard to the possibility of a causal relationship. Adverse events will be coded using the concise but valid UKU side effect rating scale [68]. We will further classify adverse events as to their severity, expectedness, and potential relatedness to the study intervention based on NIA Adverse Event and Serious Adverse Event Guidelines (September 2018). Namely, the severity of adverse events will be classified as "Mild" (being transient and requiring no therapy or evaluation), "Moderate" (of a low level of inconvenience or concern to the participant and/or the family and possibly interfering with daily activities and functioning), "Severe" (being incapacitating and requiring systemic treatment), or "Serious" (as defined in the next paragraph). Expectedness will be determined by whether the nature or severity of a given adverse event is expected to occur or not, owing to the inconsistent known information about nabilone (e.g., product monograph, case reports, or published studies). We will determine the relatedness of an event to nabilone as "Definitely Related", "Possibly Related", or "Not Related", based on a temporal relationship to the use of nabilone, as well as whether the event is unexpected or unexplained given the participant's clinical course, previous medical conditions, and concomitant medications. If nabilone is discontinued as a result of an adverse event, study personnel will document the circumstances and data leading to the discontinuation of treatment.

A serious adverse event for this study is defined as any untoward medical occurrence that is believed by the investigators to be causally related to nabilone and results in any of the following: life-threatening condition (i.e., with immediate risk of death), severe or permanent disability, or inpatient hospitalization. Serious adverse events occurring after a participant is discontinued from the study will not be reported unless the investigators consider that the event may have been caused by nabilone. An adverse event that meets the criteria for a serious adverse event between study enrollment and S-F will be reported to the CAMH Research Ethics Board (REB). Given the favorable safety and tolerability profiles of nabilone, the current principle for reporting serious adverse events to the REB conforms to the requirements for reporting Unanticipated Problems (which are unexpected, are related, or possibly related to the study intervention, and involve greater risk of harm) defined by the Canadian Association of Research Ethics Boards (CAREB) guidance document on Guidance on Reporting of Unanticipated Problems Including Adverse Events to Research Ethics Boards in Canada (July 2010).

The exploratory outcomes encompass a range of clinical and neurophysiological measures (Fig 1). Clinical measures include the CGI, ABC-I, MOAS, and General Anxiety Subscale of ADAMS to quantify participants' behavioral changes, alongside changes in caregiver stress levels estimated with the Stress Subscale of the DASS-21 and the BFDS. All are reported by the caregivers, except the CGI.

MRI scans are optional and exploratory, dependent on participants' capacity to follow the procedures and the participants' and caregivers' willingness to participate. If a given participant and his/her/their caregiver and SDM are willing and capable to undergo an MRI scan, they will receive instructions using a custom-built social script at the screening visit (S-V). The

social script comprises a series of photographs of the research procedures, environments, as well as audio files of the sounds emitted by the MRI scanner to help participants be acquainted with the scanning environment and procedures. We will use the CAMH research-dedicated 3 Tesla MRI scanner (GE MR750; General Electric, Milwaukee, WI). The acquisitions we will obtain include T1-weighted, a naturalistic film viewing functional MRI with negative emotion valence themes, and ${}^1$H MRS measuring neurometabolites including glutamate, glutamine, and myo-inositol at the anterior cingulate cortex (which is rich for CB1 and is involved in emotion regulation) scans. T1-weighted image and functional MRI will use Human Connectome Project-like data acquisition with multiband excitations [71] to reduce scan time, facilitating the successful scanning [72]. The total estimated scanning time is 35 minutes.

## Participants

**Inclusion criteria.** Participants of any sex or gender, race or ethnicity meeting all criteria listed below will be included in the study:

1. Aged ≥25 years, based on Health Canada recommendations to restrict medical cannabis use to this age [33].

2. Adults with a DSM-5 diagnosis of intellectual disabilities (ID) meeting: a. Full-scale IQ <75 on a standardized cognitive assessment reported in their prior medical record; b. A deficit in adaptive function in at least one activity of life, as estimated by the ABAS-3, as rated by the caregiver. For those whose verified records are not available, they are deemed eligible if they are recipients of Developmental Services Ontario (DSO), which is the regional access point for adult developmental disabilities-related services funded by the Ministry of Children, Community and Social Services in Ontario, Canada. All recipients of DSO services complete an eligibility process which includes reviews of cognitive and adaptive functioning using the definitions listed above. People with ID and other developmental disabilities, e.g., autism, Down syndrome, genetic conditions such as Angelman syndrome, fragile X syndrome, Prader-Willi Syndrome, etc., will also be enrolled.

3. SBPs, including aggressive, disruptive, and/or self-injurious behaviors in any situation (home, day program, clinic, etc.), as defined by a score ≥18 on the Aberrant Behavior Checklist-Irritability subscale (ABC-I), and a score ≥4 on the Clinical Global Impressions-Severity scale [61]. A consistent pattern of frequent SBPs should occur for >3 months ≥1 time per week.

4. Sexually active women of child-bearing potential must have a negative urine pregnancy test at the screening visit.

5. Sexually active women of child-bearing potential must use an effective method of birth control at least from the start of the last two normal menses before the screening visit to one month after the end of the study (i.e., completion of the safety visit). The accepted methods of contraception include total sexual abstinence if it is the usual and preferred lifestyle, or consistently and correctly taking the oral contraceptive.

6. Blood test in the previous 12 months showing liver function test with the alanine transaminase (ALT) ≤3 times the upper limit of normal and bilirubin ≤2 times the upper limit of normal.

7. At least one month since participating in another investigational drug trial.

**Exclusion criteria.** 1. History of hypersensitivity to any cannabinoid.

2. The presence of an unstable seizure disorder as defined by having not been seizure-free for at least 6 months or anticonvulsant treatment has not been stable for at least 4 weeks.

3. The presence of any clinically significant or unstable medical conditions, including cardio-vascular, liver, kidney, pulmonary disease, or the presence of known congenital brain malformation, as per investigator assessment based on medical history and chart review.

4. The presence of a lifetime diagnosis of a psychotic disorder, bipolar disorder, or substance use disorder, or current diagnosis of major depressive disorder or dementia, based on past psychiatric history noted in the medical chart, as well as Moss-PAS (ID) at S-V.

5. Family history of a psychotic disorder.

6. Change in psychotropic medications less than 4 weeks prior to study drug (nabilone) use.

7. At the time of screening, each potential participant medication list will be checked for drugs that are known to have drug-drug interactions with nabilone. The following drugs and doses will serve as exclusions:

   a. Currently on benzodiazepines at a dose more than the benzodiazepine equivalent to lorazepam 2 mg daily.

   b. Currently on medical psychostimulant, including methylphenidate ($>$100 mg daily), lisdexamfetamine ($>$70 mg daily), amphetamine/dextroamphetamine (Adderall XR®, $>$50 mg daily), dextroamphetamine (Dexedrine®, $>$50 mg daily) at a dose exceeding their respective maximum doses (as shown in the bracket after each agent) to treat attention-deficit/hyperactivity disorder (ADHD) in adults, based on the Canadian ADHD Resource Alliance (CADDRA) guideline [73].

   c. Currently on nonbenzodiazepine hypnotics, including zaleplon ($>$10 mg daily), zolpidem ($>$10 mg daily), and zopiclone ($>$7.5 mg daily), at a dose exceeding their respective suggested safety doses (as shown in the bracket after each agent), based on Canadian Recalls and Safety Alerts (https://healthycanadians.gc.ca/recall-alert-rappel-avis/).

   d. Currently on any opioid.

   e. Currently on barbiturate.

   f. Drinking any alcohol in the week prior to the screening visit.

   g. Recreational use of any psychomimetic drugs in the week prior to the screening visit, including ketamine, lysergic acid diethylamide (LSD), 3,4-methylenedioxy-methamphetamine (MDMA), magic mushrooms, phencyclidine (PCP), salvia, gamma hydroxybutyrate (GHB), bath salts, and methamphetamine.

8. Currently taking other cannabinoids, such as CBD or medical cannabis, from another source, unless participants and/or their caregivers are willing to stop such use or this treatment for at least 4 weeks prior to entering the study.

9. Participants who might travel out of the area for a significant time during the study.

10. Recently are participating in another investigational drug trial.

11. Pregnancy.

12. Sexually active women of child-bearing potential intending to give breastfeeding or get pregnant.

**Sample size and statistical analysis.**　To test Hypotheses 1 (i.e., tolerability and adherence) and 2 (i.e., safety profiles), we aim to enroll 30 patients to enter the dose titration phase, consistent with common sample sizes for Phase I trials [74].

We assume the attrition rate of 25% from the dose titration phase to entering the open-label phase. To test behavioral changes at the group level (Exploratory Objective), assuming the effect size of Cohen's d 0.6, which is similar to that detected in the study investigating the effects of nabilone on agitation associated with dementia [37], this sample size of remaining participants (N = 23) will achieve power = 0.79 with two-sided α-level set at 0.05 (G*Power 3.1).

All participants entering the dose titration phase will be included in the description of the safety profile and tolerability of nabilone (the Primary Objective). Participants completing the dose titration stage will be included in the final statistical analysis for the Exploratory Objective (i.e., exploring changes in SBPs in participants pre- to post-treatment), regardless of whether they complete the full protocol. This approach is consistent with the intention-to-treat principle.

Safety profiles will be described by the number and percentage of each adverse event in a tabular way with organ systems as the rows, and Relatedness, Expectedness, and Severity as the columns. A paired t-test will be used to investigate the pre- to post-treatment behavioral changes. Exploratory analyses will be performed to determine predictors of response including baseline individual features, using logistic regression. The behavioral and clinical data will be analyzed using IBM SPSS Statistics (Version 26). As this phase I study aims to provide preliminary signals to inform a subsequent randomized controlled trial (RCT), we will not adjust for multiple comparisons.

For MRI measures, non-parametric tests will be used to explore the brain changes with nabilone treatment, assuming that only a few participants will complete MRI scans. In addition, it is expected that the investigators will conduct further post-hoc exploratory analyses (not specified here) on these data.

## Ethics consideration, data management, and knowledge translation

This study has been approved by the CAMH Research Ethics Board (CAMH REB #135–2020) and Health Canada (Control #256517); has been registered on ClinicalTrials.gov (Identifier: NCT05273320). During the REB preparation processes, several self-advocates with neurodevelopmental disabilities and family advisors endorsed the validity and practicality of this research proposal. The language and style of the patient engagement materials, e.g., a social script, informed consent form, and acceptability/feedback questions, were edited and accommodated to be as easily accessible/readable as possible, based on the input of self-advocate and family advisors.

Participant data will be de-identified and linked with study-specific identification numbers. The data will be recorded and stored online through Research Electronic Data Capture (RED-Cap), CAMH. Data and all appropriate documentation will be stored for a minimum of 25 years after the completion of the study.

The study findings will be presented at scientific conferences and published in peer-reviewed journals. Only aggregated results with individual identifying information removed will be presented. We will also leverage the patient and family engagement resources within the Azrieli Adult Neurodevelopmental Centre, CAMH, and The Hospital for Sick Children (SickKids), Toronto to translate lessons learned from this study (e.g., for a robust recruitment plan for a larger trial) and explore how to use the information gleaned to build capacity to

carry out these types of studies in an inclusive manner. Specifically, we will highlight and disseminate the key results from this Phase I study with the families from across Canada who are linked to the Azrieli Adult Neurodevelopmental Centre, through a monthly newsletter and caregiver advisory network in the Health Care Access, Research and Developmental Disabilities (H-CARDD) Program (https://www.porticonetwork.ca/web/hcardd), and presentations at community meetings. In sum, throughout this process, we will seek input from stakeholder advisors (i.e., self-advocates with IDD, caregivers and families, via our established participatory research framework at CAMH and SickKids). Findings will be integrated and discussed with stakeholder advisors in an effort to: (i) identify meaningful ways to disseminate research findings to health service providers and stakeholders, and (ii) plan a subsequent medication clinical trial for this population in a more inclusive manner.

### Trial status

At the time of submission, recruitment has commenced.

## Discussion

This is the first study to investigate nabilone treatment for SBPs in people with IDD. The Phase I pre-pilot approach is adopted because there have yet to be any case reports or trials of nabilone with this population. Despite being theoretically promising, tolerable, and already marketed, the tolerability and safety profiles of nabilone in the IDD population must first be investigated (Phase I). The current Phase I trial is considered a pre-pilot trial to explore any signals of behavioral change with the use of nabilone, with a pilot RCT to follow.

Findings from this project will inform the design of a series of future clinical trials in the future, i.e., the next-stage Phase I/II pilot placebo-controlled RCT to test the feasibility of a large-scale RCT, followed by a Phase II/III multicentre RCT to test available options for this difficult-to-treat clinical problem [75]. If nabilone is tolerable and safe to use, we will work toward the pilot RCT and also begin to explore how nabilone impacts not only behaviour and cognition but also the neurobiological effects as investigated by brain MRI. This pharmacoimaging approach can shed light on mechanisms underpinning behavioral changes post-treatment, and enables a better understanding of response heterogeneity. Notably, if encouraging signals are detected, we acknowledge that the effect size of the identified signals may well be inflated given the small sample size and a regression to the mean effect would be expectedly in a future study [76]. Further, the current design is unable to disentangle the extent to which the time effect is conflated with the treatment effect of nabilone. Nonetheless, a substantial proportion of people with IDD have persistent and stable challenging behaviors over time [75, 77, 78], especially in the context of our stringent inclusion criteria to define the presence of SBP— we expect that this stringent definition of SBP will substantially reduce the possibility of this caveat.

The planned trial will address an evidence-practice gap in the use of cannabinoids to meet a vital need, which has been identified as a priority research area by adults with neurodevelopmental disabilities, family members and clinicians in the most recent Ontario-wide priority-setting exercise initiative (https://braininstitute.ca/programs-opportunities/setting-research-priorities/neurodevelopmental-disorders-psp). Anecdotally, some adults with IDD are already trialling cannabinoids in different ways, as an alternative to commonly used psychotropic medications, with scarce systematic studies, cf. [31]. A national psychoeducational course on mental health and IDD with families would suggest that these families are very concerned about medication overuse (unpublished data). Nonetheless, this population has long been excluded from clinical trials [79], despite concerns about the current standard of clinical

practices. This exclusion is an inequity problem and limits this population's access to the development of innovative interventions and supports. Thus, targeted and inclusive clinical trials, such as our proposal here, will be the start of an endeavour to directly address health inequity experienced by individuals with IDD and their families. The findings and experiences earned, in combination with the stakeholders' inputs, will facilitate more accessible, inclusive, and ethical clinical trial designs for people with IDD.

## Supporting information

**S1 File. The exampled informed consent form.**
(PDF)

**S2 File. The exampled assent form.**
(PDF)

**S3 File. The custom feedback questionnaire.**
(PDF)

**S4 File. Spirit checklist.**
(PDF)

**S5 File.**
(PDF)

## Acknowledgments

We thank the self-advocates and family advisors at the Azrieli Adult Neurodevelopmental Centre, Centre for Addiction and Mental Health, Toronto, Ontario, Canada, who helped us evaluate the feasibility of the design of this study and edit the language in the informed consent form and assent form to facilitate the readability of these documents for people with IDD.

## Author Contributions

**Conceptualization:** Hsiang-Yuan Lin, Yona Lunsky, Tarek K. Rajji.

**Data curation:** Hsiang-Yuan Lin, Pushpal Desarkar, Stephanie H. Ameis, Meng-Chuan Lai.

**Formal analysis:** Hsiang-Yuan Lin, Wei Wang.

**Funding acquisition:** Hsiang-Yuan Lin, Elia Abi-Jaoude.

**Investigation:** Hsiang-Yuan Lin.

**Methodology:** Hsiang-Yuan Lin, Elia Abi-Jaoude.

**Project administration:** Hsiang-Yuan Lin.

**Resources:** Hsiang-Yuan Lin, Pushpal Desarkar, Yona Lunsky, Tarek K. Rajji.

**Software:** Hsiang-Yuan Lin.

**Supervision:** Yona Lunsky, Tarek K. Rajji.

**Validation:** Hsiang-Yuan Lin.

**Visualization:** Hsiang-Yuan Lin.

**Writing – original draft:** Hsiang-Yuan Lin.

**Writing – review & editing:** Elia Abi-Jaoude, Pushpal Desarkar, Wei Wang, Stephanie H. Ameis, Meng-Chuan Lai, Yona Lunsky, Tarek K. Rajji.

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
