## [Decision Letter · Decision Letter 0]

27 Dec 2022

PONE-D-22-30722Nabilone treatment for severe behavioral problems in adults with intellectual and developmental disabilities: protocol for a phase I open-label clinical trialPLOS ONE

Dear Dr. Lin,

Thank you for submitting your manuscript to PLOS ONE. After careful consideration, we feel that it has merit but does not fully meet PLOS ONE’s publication criteria as it currently stands. Therefore, we invite you to submit a revised version of the manuscript that addresses the points raised during the review process. The comments from the 2 reviewers are straightforward, and you should have no problems in addressing them. The revised paper may make a nice contribution to the literature, especially in the field of IDD.

We look forward to receiving your revised manuscript.

Kind regards,

Robert Didden

Academic Editor

PLOS ONE

Journal Requirements:

"This project is funded by the CAMH Innovation Fund of the Alternative Funding Plan for the Academic Health Sciences Centres of Ontario, and the University of Toronto Department of Psychiatry Excellence Funds. H-YL is supported by is supported by the Azrieli Adult Neurodevelopmental Centre at Centre for Addiction and Mental Health, and an Academic Scholar Award from the Department of Psychiatry, University of Toronto. The funders had no role in the design of the study; in the collection, analyses, or interpretation of data; in the writing of the manuscript, or in the decision to publish the protocol."

"This project is funded by the CAMH Innovation Fund of the Alternative Funding Plan for the Academic Health Sciences Centres of Ontario, and the University of Toronto Department of Psychiatry Excellence Funds. H-YL is supported by is supported by the Azrieli Adult Neurodevelopmental Centre at Centre for Addiction and Mental Health, and an Academic Scholar Award from the Department of Psychiatry, University of Toronto. The funders had no role in the design of the study; in the collection, analyses, or interpretation of data; in the writing of the manuscript, or in the decision to publish the protocol."

Reviewers' comments:

Reviewer's Responses to Questions

**Comments to the Author**

1. Does the manuscript provide a valid rationale for the proposed study, with clearly identified and justified research questions?

Reviewer #1: Yes

Reviewer #2: Yes

2. Is the protocol technically sound and planned in a manner that will lead to a meaningful outcome and allow testing the stated hypotheses?

Reviewer #1: Yes

Reviewer #2: Yes

3. Is the methodology feasible and described in sufficient detail to allow the work to be replicable?

Reviewer #1: Yes

Reviewer #2: Yes

4. Have the authors described where all data underlying the findings will be made available when the study is complete?

Reviewer #1: No

Reviewer #2: Yes

5. Is the manuscript presented in an intelligible fashion and written in standard English?

Reviewer #1: Yes

Reviewer #2: Yes

6. Review Comments to the Author

You may also provide optional suggestions and comments to authors that they might find helpful in planning their study.

Reviewer #1: Generally, the protocol appears fairly thorough and should yield useful data.

Since the safety assessments are an important part of this protocol, albeit the adverse event profile is expected to be good, please provide an additional paragraph or two under the assessments which is more in keeping with industry standards describing these assessments.

Please cover the coding to be used and how the various types of adverse events will be defined. Is there any requirement for reporting adverse events to a regulatory or oversight body in this trial? If so, please detail these requirements and their implementation.

For analysis of the adverse event data, please plan to produce industry standard adverse event tables, typically consisting of overall adverse events, serious adverse events, attribution to study drug, and tabulation by system and organ class. If other types of adverse events are planned to be defined, such as serious adverse events or unexpected adverse events, please describe these.

Tolerability will be declared if 80% of the participants complete the study protocol. Does completion include the tapering phase? This phase is not necessarily related to ongoing treatment with nabilone, but it is also reasonable to include this phase as representing all phases of treatment. What will be the definition of completing the study protocol?

The statistical analysis appears to be based on paired t-tests. This is not unreasonable, though the hypothesized power will depend somewhat on the actual distributions under consideration. Will any adjustments for multiple testing be made? If not, please explain briefly, and if so, please describe the planned adjustments.

Since this study will not be placebo-controlled, it should be noted that if resulting effects are chosen on the basis of their size, that they will almost certainly be overinflated and one could expect a regression to the mean effect in a future study.

Also, the results from this study will be conflated with any effects that would manifest over time in this population after selection. It would be best if the authors can show that any such effects are either non-existent or expected to be minimal.

If the data management software and statistical analysis software to be used are known, please describe them in the statistical methods section.

Reviewer #2: This is a very good quality study protocol.

I have only 2 minor remarks that should be addressed.

1. In line 150, the authors claim that current evidence suggests that nabilone could be promising - however, from the manuscript this is not really clear. Pleas explain this more in detail.

2. In the manuscript it is mentioned that one of the goals is to assess adherence, however this is not mentioned in the abstract and not pointed out in the manuscript.

7. PLOS authors have the option to publish the peer review history of their article (what does this mean?). If published, this will include your full peer review and any attached files.

Reviewer #1: No

Reviewer #2: No

---

## [Author Response · Author response to Decision Letter 0]

28 Jan 2023

Reviewer #1: 

General comment. Generally, the protocol appears fairly thorough and should yield useful data.

Re: Thanks for your positive appraisal and constructive suggestions. We have addressed each of your comments as follows. 

R1.1. Since the safety assessments are an important part of this protocol, albeit the adverse event profile is expected to be good, please provide an additional paragraph or two under the assessments which is more in keeping with industry standards describing these assessments.

R1.2. Please cover the coding to be used and how the various types of adverse events will be defined. Is there any requirement for reporting adverse events to a regulatory or oversight body in this trial? If so, please detail these requirements and their implementation.

R1.3. For analysis of the adverse event data, please plan to produce industry standard adverse event tables, typically consisting of overall adverse events, serious adverse events, attribution to study drug, and tabulation by system and organ class. If other types of adverse events are planned to be defined, such as serious adverse events or unexpected adverse events, please describe these.

Re: We appreciate your comments regarding the adverse events (Comments 1-3). As these 3 comments are highly relevant, we decided to respond to them together here. We have added two paragraphs in the subsection of Outcome Measures (Page 15 Line 323 to Page 17 Line 353): 

“ An adverse event will be defined as any untoward medical occurrence in a participant, temporally associated with his/her/their involvement in this study (from the time that the informed consent document is signed to the S-F), without regard to the possibility of a causal relationship. Adverse events will be coded using the concise but valid UKU side effect rating scale (68). We will further classify adverse events as to their severity, expectedness, and potential relatedness to the study intervention based on NIA Adverse Event and Serious Adverse Event Guidelines (September 2018). Namely, the severity of adverse events will be classified as “Mild” (being transient and requiring no therapy or evaluation), “Moderate” (of a low level of inconvenience or concern to the participant and/or the family and possibly interfering with daily activities and functioning), “Severe” (being incapacitating and requiring systemic treatment), or “Serious” (as defined in the next paragraph). Expectedness will be determined by whether the nature or severity of a given adverse event is expected to occur or not, owing to the inconsistent known information about nabilone (e.g., product monograph, case reports, or published studies). We will determine the relatedness of an event to nabilone as “Definitely Related”, “Possibly Related”, or “Not Related”, based on a temporal relationship to the use of nabilone, as well as whether the event is unexpected or unexplained given the participant’s clinical course, previous medical conditions, and concomitant medications. If nabilone is discontinued as a result of an adverse event, study personnel will document the circumstances and data leading to the discontinuation of treatment. 

 A serious adverse event for this study is defined as any untoward medical occurrence that is believed by the investigators to be causally related to nabilone and results in any of the following: life-threatening condition (i.e., with immediate risk of death), severe or permanent disability, or inpatient hospitalization. Serious adverse events occurring after a participant is discontinued from the study will not be reported unless the investigators consider that the event may have been caused by nabilone. An adverse event that meets the criteria for a serious adverse event between study enrollment and S-F will be reported to the CAMH Research Ethics Board (REB). Given the favorable safety and tolerability profiles of nabilone, the current principle for reporting serious adverse events to the REB conforms to the requirements for reporting Unanticipated Problems (which are unexpected, are related, or possibly related to the study intervention, and involve greater risk of harm) defined by the Canadian Association of Research Ethics Boards (CAREB) guidance document on Guidance on Reporting of Unanticipated Problems Including Adverse Events to Research Ethics Boards in Canada (July 2010).” 

Following your suggestion, in these two paragraphs, we also highlighted the coding and classification of adverse events.

Further, in the subsection of Statistical Analysis, we added a tabular presentation of the adverse events as per your suggestion. Please see Page 23 Lines 479-481: 

“Safety profiles will be described by the number and percentage of each adverse event in a tabular way with organ systems as the rows, and Relatedness, Expectedness, and Severity as the columns. ” 

R1.4. Tolerability will be declared if 80% of the participants complete the study protocol. Does completion include the tapering phase? This phase is not necessarily related to ongoing treatment with nabilone, but it is also reasonable to include this phase as representing all phases of treatment. What will be the definition of completing the study protocol?

Re: Thank you for giving us this opportunity to clarify this point. We defined tolerability by the completion of the open-label treatment stage because tolerability focuses on whether the participants can tolerate the study drug and the related adverse events. This definition has been added to the study objectives on Page 7 Line 154.

“ To evaluate tolerability, we hypothesize that 80% of participants will complete the open-label treatment stage.”

R1. 5. The statistical analysis appears to be based on paired t-tests. This is not unreasonable, though the hypothesized power will depend somewhat on the actual distributions under consideration. Will any adjustments for multiple testing be made? If not, please explain briefly, and if so, please describe the planned adjustments.

Re: As this phase 1 study aims to provide preliminary signals to inform a subsequent randomized controlled trial, we will not adjust for multiple comparisons. Please see Page 23 Line 484 to Page 24 Line 486 :

“As this phase 1 study aims to provide preliminary signals to inform a subsequent randomized controlled trial (RCT), we will not adjust for multiple comparisons.”

R1. 6. Since this study will not be placebo-controlled, it should be noted that if resulting effects are chosen on the basis of their size, that they will almost certainly be overinflated and one could expect a regression to the mean effect in a future study.

R1. 7. Also, the results from this study will be conflated with any effects that would manifest over time in this population after selection. It would be best if the authors can show that any such effects are either non-existent or expected to be minimal.

Re: Thank you for raising these two important points on the interpretation of results. As these two comments are highly relevant, we decided to respond to them together. We totally agree with your insight on potential regression to the mean. Also, we agree that our findings may be conflated with the time effect given the open-label nature. However, a substantial proportion of people with IDD have persistent and stable challenging behaviors over time, especially in the context of our stringent inclusion criteria to define the presence of SBP. We believe the way SBPs are defined in this study could substantially reduce this caveat. We have highlighted these important considerations surrounding data interpretation in the Discussion. Please see Page 26 Lines 541-548:

“Notably, if encouraging signals are detected, we acknowledge that the effect size of the identified signals may well be inflated given the small sample size and a regression to the mean effect would be expectedly in a future study (76). Further, the current design is unable to disentangle the extent to which the time effect is conflated with the treatment effect of nabilone. Nonetheless, a substantial proportion of people with IDD have persistent and stable challenging behaviors over time (75, 77, 78), especially in the context of our stringent inclusion criteria to define the presence of SBP—we expect that this stringent definition of SBP will substantially reduce the possibility of this caveat.”

References: 

Dimian AF, Symons FJ. A systematic review of risk for the development and persistence of self-injurious behavior in intellectual and developmental disabilities. Clin Psychol Rev. 2022;94:102158.

Totsika V, Hastings RP. Persistent challenging behaviour in people with an intellectual disability. Curr Opin Psychiatry. 2009;22(5):437-41.

Totsika V, Toogood S, Hastings RP, Lewis S. Persistence of challenging behaviours in adults with intellectual disability over a period of 11 years. J Intellect Disabil Res. 2008;52(Pt 5):446-57.

R1. 8. If the data management software and statistical analysis software to be used are known, please describe them in the statistical methods section.

Re: We have specified this detail on Page 23 Lines 483 and 484:

“The behavioral and clinical data will be analyzed using IBM SPSS Statistics (Version 26). “ 

Reviewer #2: 

R2.1. In line 150, the authors claim that current evidence suggests that nabilone could be promising - however, from the manuscript this is not really clear. Please explain this more in detail.

Re: Thank you for this comment. We have added the rationale to justify this statement (Page 7 Lines 147 and 148):

“Based on the encouraging evidence in patients with dementia (37) and Parkinson’s disease (40) as well as anecdotal clinical observations, nabilone could be a promising and novel treatment for SBPs in adults with IDD.”

R2.2. In the manuscript it is mentioned that one of the goals is to assess adherence, however this is not mentioned in the abstract and not pointed out in the manuscript.

Re: We thank the reviewer for bringing this point to our attention. We have mentioned the adherence as one of the objectives in Abstract (Page 2 Line 46):

“Thus, we propose this first-ever Phase I pre-pilot open-label clinical trial to obtain preliminary data on the adherence, tolerability and safety profiles of nabilone in adults with IDD”

In addition, the method of assessing adherence was originally stated in the subsection of Outcome Measures (Page 15 Line 321):

“The primary outcome measures are adherence (as quantified by counting pill usage)”

---

## [Decision Letter · Decision Letter 1]

8 Feb 2023

Nabilone treatment for severe behavioral problems in adults with intellectual and developmental disabilities: protocol for a phase I open-label clinical trial

PONE-D-22-30722R1

Dear Dr. Lin,

We’re pleased to inform you that your manuscript has been judged scientifically suitable for publication and will be formally accepted for publication once it meets all outstanding technical requirements.

Kind regards,

Robert Didden

Academic Editor

PLOS ONE

Additional Editor Comments (optional):

Reviewers' comments:

Reviewer's Responses to Questions

**Comments to the Author**

1. Does the manuscript provide a valid rationale for the proposed study, with clearly identified and justified research questions?

Reviewer #1: Yes

2. Is the protocol technically sound and planned in a manner that will lead to a meaningful outcome and allow testing the stated hypotheses?

Reviewer #1: Yes

3. Is the methodology feasible and described in sufficient detail to allow the work to be replicable?

Reviewer #1: Yes

4. Have the authors described where all data underlying the findings will be made available when the study is complete?

Reviewer #1: Yes

5. Is the manuscript presented in an intelligible fashion and written in standard English?

Reviewer #1: Yes

6. Review Comments to the Author

You may also provide optional suggestions and comments to authors that they might find helpful in planning their study.

Reviewer #1: This is a review of a revised manuscript. All issues have been dealt with in the authors' revision.

7. PLOS authors have the option to publish the peer review history of their article (what does this mean?). If published, this will include your full peer review and any attached files.

Reviewer #1: No

---

## [Editor Report · Acceptance letter]

3 Apr 2023

PONE-D-22-30722R1 

Nabilone treatment for severe behavioral problems in adults with intellectual and developmental disabilities: protocol for a phase I open-label clinical trial 

Dear Dr. Lin:

I'm pleased to inform you that your manuscript has been deemed suitable for publication in PLOS ONE. Congratulations! Your manuscript is now with our production department. 

Kind regards, 

on behalf of

Professor Robert Didden 

Academic Editor

PLOS ONE